# Exploring the Combined Power of Covariance and Hessian Matrices Eigenanalysis for Binary Classification

## Abstract

Covariance and Hessian matrices have been analyzed separately in the literature for classification problems. However, integrating these matrices has the potential to enhance their combined power in improving classification performance. We present a novel approach that combines the eigenanalysis of a covariance matrix evaluated on a training set with a Hessian matrix evaluated on a deep learning model to achieve optimal class separability in binary classification tasks. Our approach is substantiated by formal proofs that establish its capability to maximize between-class mean distance and minimize within-class variances. By projecting data into the combined space of the most relevant eigendirections from both matrices, we achieve optimal class separability as per the linear discriminant analysis (LDA) criteria. Empirical validation across neural and health datasets consistently supports our theoretical framework and demonstrates that our method outperforms established methods. Our method stands out by addressing both LDA criteria, unlike PCA and the Hessian method, which predominantly emphasize one criterion each. This comprehensive approach captures intricate patterns and relationships, enhancing classification performance. Furthermore, through the utilization of both LDA criteria, our method outperforms LDA itself by leveraging higher-dimensional feature spaces, in accordance with Cover's theorem, which favors linear separability in higher dimensions. Our method also surpasses kernel-based methods and manifold learning techniques in performance. Additionally, our approach sheds light on complex DNN decision-making, rendering them comprehensible within a 2D space.

## 1 Introduction

Binary classification is a fundamental task in machine learning, where the goal is to assign data points to one of two classes. The accuracy and effectiveness of binary classifiers depend on their ability to separate the two classes accurately. However, achieving optimal class separability can be challenging, especially when dealing with complex and high-dimensional data.

Traditional approaches often rely on analyzing either the covariance matrix (Nagai, 2020; Minh & Murino, 2017; Serra et al., 2014; Lenc & Vedaldi, 2016; Hoff & Niu, 2011; Kuo & Landgrebe, 2002; Lam, 2019) or the Hessian matrix (Dawid et al., 2021; Fu et al., 2020; Yao et al., 2019; Krishnasamy & Paramesran, 2016; Wiesler et al., 2013; Byrd et al., 2011; Martens, 2010) separately to optimize machine learning models. In the field of Evolution Strategies (ESs), a recent work (Shir & Yehudayoff, 2020) explored the relationship between the covariance matrix and the landscape Hessian, highlighting the statistical learning capabilities of ESs using isotropic Gaussian mutations and rank-based selection. While this study provides valuable insights into ESs' learning behavior, it does not investigate the practical integration of the two matrices.

When analyzing the covariance matrix, the focus is on capturing the inherent patterns of variability within the data, allowing for a compact representation that highlights relationships between different dimensions. On the other hand, the analysis of the Hessian matrix aims to find the direction in which the classes are best separated, by maximizing the curvature along the discriminative directions.

However, these separate analyses fail to fully leverage the synergistic effects that can arise from integrating the information contained in both matrices.

To tackle this challenge, we present a novel approach that combines the eigenanalysis of the covariance matrix evaluated on a training set with the Hessian matrix evaluated on a deep learning model. By integrating these matrices, our method aims to optimize class separability by simultaneously maximizing between-class mean distance and minimizing within-class variances, which are the fundamental criteria of linear discriminant analysis (LDA) (Fisher, 1936; Xanthopoulos et al., 2013). By utilizing these criteria, we leverage the foundational principles that have been established and proven over decades of LDA's application in various fields like medicine (Sharma et al., 2012; Sharma & Paliwal, 2008; Moghaddam et al., 2006; Dudoit et al., 2002; Chan et al., 1995), agriculture (Tharwat et al., 2017; Gaber et al., 2015; Rezzi et al., 2005; Héberger et al., 2003; Chen et al., 1998), and biometrics (Paliwal & Sharma, 2012; Yuan & Mu, 2007; Park & Park, 2005; Wang & Tang, 2004; Yu & Yang, 2001; Chen et al., 2000; Haeb-Umbach & Ney, 1992). LDA has been demonstrating the reliability and usefulness of the two criteria as indicators of discriminative power. Thus, by building upon this well-established foundation, our approach will inherit the strengths and reliability that have been demonstrated by LDA. This integrated approach holds the promise of enhancing classification performance beyond conventional methods that treat the two matrices separately.

## 2 METHODOLOGY

### 2.1 APPROACH: COMBINING EIGENANALYSIS OF COVARIANCE AND HESSIAN MATRICES

In this subsection, we describe our novel approach for combining the eigenanalysis of the covariance matrix and the Hessian matrix to achieve optimal class separability in binary classification tasks.

1. Covariance matrix eigenanalysis: A covariance matrix $\text{Cov}(\boldsymbol{\theta})$ is a $D \times D$ matrix, where $D$ represents the dimensionality of the predictor attributes. Each element of the covariance matrix reflects the covariance between the two predictor attributes $\theta_1$ and $\theta_2$:

$$\text{Cov}(\theta_1, \theta_2) = \frac{1}{n-1} \sum_{i=1}^{n} (\theta_{1i} - \bar{\theta}_1)(\theta_{2i} - \bar{\theta}_2)$$

where $\theta_{1i}$ and $\theta_{2i}$ are the corresponding observations of these predictor attributes for the $i$-th data instance, $\bar{\theta}_1$ and $\bar{\theta}_2$ are the sample means of the predictor attributes $\theta_1$ and $\theta_2$, respectively, and $n$ is the number of data instances or observations.

Performing eigenanalysis on the covariance matrix $\text{Cov}(\boldsymbol{\theta})$, we obtain eigenvalues $\lambda_i$ and corresponding eigenvectors $\mathbf{v}_i$. The leading eigenvector $\mathbf{v}_1$ associated with the largest eigenvalue $\lambda_1$, as captured in the eigen-equation $\text{Cov}(\boldsymbol{\theta}) \cdot \mathbf{v}_1 = \lambda_1 \cdot \mathbf{v}_1$, represents the principal direction with the highest variance.

2. Hessian matrix eigenanalysis: Next, we compute a Hessian matrix evaluated on a deep learning model trained on the same training set. We employ a deep neural network (DNN) architecture consisting of four fully connected layers, each followed by a Rectified Linear Unit (ReLU) activation. The final layer employs a sigmoid activation function to yield a probability value within the range of 0 to 1. During training, we utilize the binary cross-entropy loss function. The binary cross-entropy loss is given by

$$\text{BCELoss} = -\sum_{i=1}^{n} \log p_\theta(c_i \mid x_i),$$

where $c_i$ is the actual class for data instance $x_i$. The probability $p_\theta(c_i \mid x_i)$ is the predicted probability of instance $x_i$ belonging to class $c_i$, as estimated by the DNN model parameterized by $\theta$.

The Hessian of the loss function is then:

$$H_\theta = \nabla_\theta^2 \text{BCELoss} = \nabla_\theta^2 \left[ -\sum_{i=1}^{n} \log p_\theta(c_i \mid x_i) \right] = -\sum_{i=1}^{n} \left[ \nabla_\theta^2 \log p_\theta(c_i \mid x_i) \right].$$

Performing eigenanalysis on the Hessian matrix $H_{\boldsymbol{\theta}}$, we obtain eigenvalues $\lambda_i'$ and corresponding eigenvectors $\mathbf{v}_i'$. The leading eigenvector $\mathbf{v}_1'$ associated with the largest eigenvalue $\lambda_1'$, as captured in the eigen-equation $H_{\boldsymbol{\theta}} \cdot \mathbf{v}_1' = \lambda_1' \cdot \mathbf{v}_1'$, represents the direction corresponding to the sharpest curvature.

3. Integration of matrices and projection of data:

   To combine the power of covariance and Hessian matrices, we project the data into the combined space of the most relevant eigendirections. Let $\mathbf{U}$ be the matrix containing the leading eigenvectors from both matrices:

$$\mathbf{U} = [\mathbf{v}_1, \mathbf{v}_1'] \tag{1}$$

   The 2D projection of the data, denoted as $\mathbf{X}_{\text{proj}}$, is obtained by:

$$\mathbf{X}_{\text{proj}} = \mathbf{X} \cdot \mathbf{U} \tag{2}$$

   Here, $\mathbf{X}$ is the original data matrix, and $\mathbf{X}_{\text{proj}}$ represents the final output of the proposed method—a 2D projection capturing both statistical spread and discriminative regions of the data.

   Unlike LDA, which aims to optimize both criteria simultaneously along a single direction for binary classification, constrained by the limitation that the number of linear discriminants is at most $c - 1$ where $c$ is the number of class labels Ye et al. (2004), our approach provides more flexibility and control. By working on two separate directions, we specifically focus on minimizing the within-class variances in one direction while maximizing the between-class mean distance in the other direction.

## 2.2 FORMAL FOUNDATION: MAXIMIZING THE SQUARED BETWEEN-CLASS MEAN DISTANCE AND MINIMIZING THE WITHIN-CLASS VARIANCE

In this subsection, we establish two theorems along with their respective proof sketches that form the theoretical basis for our approach. Full proofs for both theorems are available in Appendix A.

### 2.2.1 THEOREM 1: MAXIMIZING COVARIANCE FOR MAXIMIZING SQUARED BETWEEN-CLASS MEAN DISTANCE.

Consider two sets of 1D data points representing two classes, denoted as $C_1$ and $C_2$, each consisting of $n$ samples. The data in $C_1$ and $C_2$ follow the same underlying distribution centered around their respective means, $m_1$ and $m_2$. Here, $C_2$ can be understood as a reflection of $C_1$ with the axis of reflection positioned at the overall mean, denoted as $m$. Furthermore, the variances of $C_1$ and $C_2$, are equal, denoted as $s_1^2 = s_2^2 = s_w^2$, and the combined data from $C_1$ and $C_2$ has a variance of $s^2$. The between-class mean distance, denoted as $d$, represents the separation between the means of $C_1$ and $C_2$. We establish the following relationship:

$$s^2 = \frac{d^2}{4(1 - \lambda)} \tag{3}$$

where $\lambda = \frac{s_w^2}{s^2}$ signifies a constant between 0 and 1 that reflects the distribution of the original data when $C_1$ and $C_2$ are considered as projected representations.

**Proof sketch:**

1. Consider two classes $C_1$ and $C_2$ with identical underlying distributions.

2. Simplify the expression for the combined data variance $s^2$ to $s_w^2 + \frac{1}{4}d^2$.

3. Apply the Variance Ratio Preservation Theorem for Projection onto a Vector to relate $s_w^2$ and $s^2$ through $\lambda$.

4. Simplify the expression to obtain $s^2 = \frac{d^2}{4(1-\lambda)}$.

5. Conclude that maximizing $s^2$ maximizes $d^2$, fulfilling the theorem's objective.

### 2.2.2 THEOREM 2: MAXIMIZING HESSIAN FOR MINIMIZING WITHIN-CLASS VARIANCE

Let $\theta$ be a parameter of the model, and $H_\theta$ denote the Hessian of the binary cross-entropy loss function with respect to $\theta$. We define the within-class variance as the variance of a posterior distribution $p_\theta(\theta \mid c_i)$, which represents the distribution of the parameter $\theta$ given a class $c_i$. We denote the variance of this posterior distribution as $\sigma^2_{post}$. We establish the following relationship:

$$H_\theta = \frac{1}{\sigma^2_{\text{post}}}. \tag{4}$$

**Proof sketch:**

1. Consider the parameter $\theta$ and the Hessian $H_\theta$ of the binary cross-entropy loss function. Define the within-class variance $\sigma^2_{\text{post}}$ as the variance of the posterior distribution $p_\theta(\theta \mid c_i)$.

2. Approximate the Hessian $H_\theta$ as the Fisher information using the expectation of the squared gradient of the log-likelihood (Barshan et al., 2020).

3. Assume a known normal likelihood distribution for $p_\theta(c_i \mid \theta)$ with mean $\mu$ and standard deviation $\sigma$. Compute the Fisher information as $\frac{1}{\sigma^2}$.

4. Considering a uniform prior distribution $p(\theta)$ within the plausible range of $\theta$ and the evidence $p(c_i)$ as a known constant, apply Bayes' formula to derive that $\sigma^2_{\text{post}} = \sigma^2$.

5. Derive $H_\theta = \frac{1}{\sigma^2_{\text{post}}}$.

6. Conclude that maximizing $H_\theta$ minimizes $\sigma^2_{\text{post}}$, achieving the theorem's objective.

Theorem 1 and Theorem 2 respectively suggest that maximizing the variance of projected data and the Hessian effectively maximize squared between-class mean distance and minimize within-class variances. These theorems provide the theoretical foundation for the eigenanalysis of the covariance and Hessian matrices, crucial steps in our proposed method for improving class separability based on the two LDA criteria.

## 3 COMPLEXITY ANALYSIS

In assessing the computational demands of our proposed method, we employ big O notation to describe the worst-case time complexity. The overall time complexity of our method encompasses several tasks, including covariance and Hessian matrix computations ($\mathcal{O}(N \cdot F^2)$), eigenanalysis of these matrices ($\mathcal{O}(F^3)$), selection of eigenvectors corresponding to leading eigenvalues ($\mathcal{O}(1)$), and data projection into the combined space ($\mathcal{O}(N \cdot F)$). Here, $N$ signifies the number of data points in the training set, while $F$ represents the number of features per data point. It is essential to note that this analysis considers our method's computational demands under the assumption that a pre-existing DNN is in place.

Comparatively, the time complexity of LDA is primarily influenced by the calculation of both within-class and between-class scatter matrices ($\mathcal{O}(N \cdot F^2)$) and subsequent eigenanalysis ($\mathcal{O}(F^3)$). In both methods, the dominant complexity factor is either the matrix computations ($\mathcal{O}(N \cdot F^2)$) when there are more examples than features or the eigenanalysis step ($\mathcal{O}(F^3)$) otherwise.

This analysis reveals that our proposed method exhibits a comparable computational profile to the method under improvement. Therefore, our approach offers enhanced class separability and interpretability without significantly increasing computational demands, making it a practical choice for real-world applications.

## 4 EXPERIMENT

### 4.1 ASSESSED DIMENSIONALITY REDUCTION TECHNIQUES

In our evaluation, we compare nine distinct dimensionality reduction and data projection techniques, each offering unique insights: principal component analysis (PCA), kernel PCA (KPCA) (Schölkopf

et al., 1997), Hessian, uniform manifold approximation and projection (UMAP) (McInnes et al., 2018), locally linear embedding (LLE) (Roweis & Saul, 2000), linear optimal low-rank projection (LOL) (Vogelstein et al., 2021), linear discriminant analysis (LDA), kernel discriminant analysis (KDA) (Mika et al., 1999), and the proposed approach. PCA involves projection onto the primary two covariance eigenvectors, i.e., it applies KPCA with a linear kernel. Similarly, LDA employs KDA with a linear kernel. The Hessian method projects data onto the leading two Hessian eigenvectors. Notably, for both KPCA and KDA, the best kernels (linear, polynomial, RBF, sigmoid, or cosine similarity) and their associated parameters (kernel coefficient or degree) are determined using grid search tailored to each dataset. The proposed method combines the most relevant eigenvectors derived from both the covariance and Hessian matrices, as specified by Eqs(1, 2).

## 4.2 CLASS SEPARABILITY ASSESSMENT VIA LINEAR SVMS

The nine dimensionality reduction methods we assess are designed to transform high-dimensional data into more manageable 1D or 2D spaces while simultaneously enhancing or preserving class separability. Linear SVMs enable us to create decision boundaries that are readily visualized in these 1D or 2D spaces. Thus, we utilize SVMs with a linear kernel to evaluate and visualize the extent of class separability achieved through the projections.

In addition to the visualization of decision boundaries, we employ a comprehensive set of evaluation metrics to quantify the performance of the dimensionality reduction methods. These metrics include the F1 score, which measures the balance between precision and recall, the Area Under the Receiver Operating Characteristic Curve (AUC ROC), which assesses the model's ability to distinguish between positive and negative classes, and Cohen's kappa, a statistic that gauges the agreement between the predicted and actual class labels.

## 4.3 DATASETS

We will conduct our assessment of the nine distinct methods on the following datasets:

**Widely recognized benchmark datasets.** We evaluate our approach using three widely recognized benchmark datasets for binary classification: the Wisconsin breast cancer dataset (Street et al., 1993), the heart disease dataset (Detrano et al., 1989), and the Pima Indians diabetes dataset (Smith et al., 1988). Prior to applying various dimensionality reduction methods, we enact standard data preprocessing techniques on the original datasets, including handling of missing data, one-hot encoding for categorical variables, and data normalization.

**Neural spike train dataset.** A spike train is a sequence of action potentials (spikes) emitted by a neuron over time. The neural spike train dataset used in this research consists of recordings from rat's neurons during drug application from a multi-electrode array (Tsai et al., 2015; Heuschkel et al., 2002). The data, comprising 221 records, represents the final dataset after all preprocessing steps suggested by (Lazarevich et al., 2023). Each record contains 15 time-series features extracted using the 'tsfresh' package (Christ et al., 2018) and 1 class attribute indicating whether the neuron is non-responsive (0) or responsive (1). Exploratory data analysis revealed an imbalance in the dataset, with 190 non-responsive neurons (86%) and 31 responsive neurons (14%).

To provide an unbiased assessment of our method's performance, we conduct extensive experiments on the datasets with 10-fold cross-validation with the exception of the neural spike train dataset. Due to the limited number of positive cases in the neural spike train dataset, we performed 5-fold cross-validation to ensure a reasonable sample size for validation.

## 4.4 RESULTS

The results depicted in Figure 1, which pertain to the WBCD dataset, along with the corresponding findings presented in Appendix B for the other three datasets, collectively reaffirm the consistency and robustness of our proposed approach across diverse datasets. The heatmaps shown in Figure 1(b) and Appendix B consistently demonstrate the reduction in squared between-class mean distance as more covariance eigenvectors are incorporated, aligning with our established theoretical framework (Eq(3)). Furthermore, Figure 1(c) and its counterparts in the appendix reveal the ascending order of within-class variances, in sync with the descending order of the Hessian, supporting our theo-

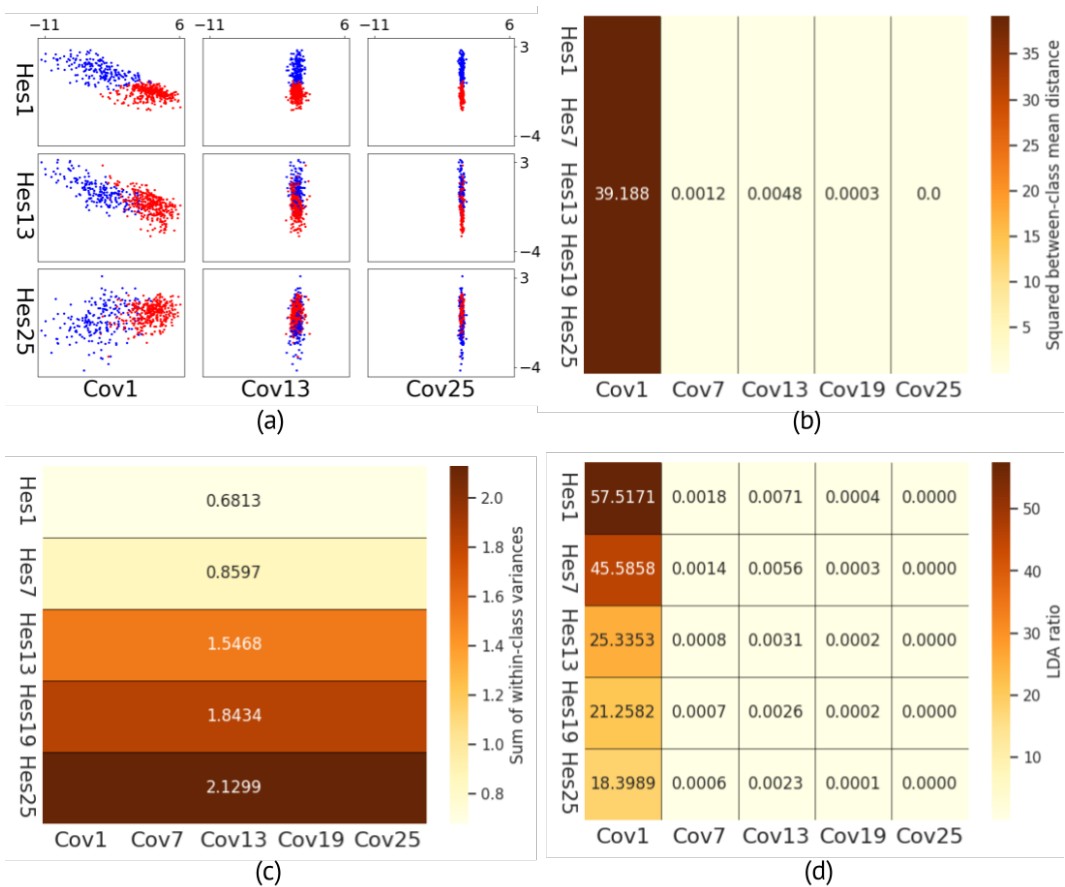

Figure 1: **Projection of the Wisconsin breast cancer data into different combined spaces of the covariance and Hessian eigenvectors.** **(a)** Nine selected projection plots, each representing data projected onto a distinct space created by combining the first three covariance and first three Hessian eigenvectors. **(b)** Heatmap showing the squared between-class mean distance for projections onto varying combinations of covariance and Hessian eigenvectors. The heatmap demonstrates that the values remain constant vertically across different Hessian eigenvectors, while exhibiting a noticeable descending order horizontally, aligning with the descending order of the variance. These results essentially concretize our formal premise, empirically validating the linear relationship described in Eq(3) between the variance and the squared between-class mean distance. **(c)** Heatmap showing the sum of within-class variances for projections onto different combinations of covariance and Hessian eigenvectors. The values remain constant horizontally across different covariance eigenvectors but exhibit a clear ascending order vertically, aligning with the descending order of the Hessian. These empirical results validate the negative correlation between the Hessian and the within-class variance described in Eq(4) within the framework of our theoretical foundation. **(d)** Heatmap displaying the LDA ratio, representing the ratio between the squared between-class mean distances presented in (b) and the corresponding within-class variances shown in (c). The highest LDA ratio is observed for the combination of the first Hessian eigenvector with the first covariance eigenvector. Notably, a general descending pattern is observed both horizontally and vertically across different combinations, indicating that both covariance eigenanalysis (represented along the horizontal direction) and Hessian eigenanalysis (represented along the vertical direction) equally contribute to the class separability (represented by the LDA ratio).

retical foundation (Eq(4)). Furthermore, Figure 1(d) and its counterparts illustrate the LDA ratio, emphasizing the equal contributions of covariance and Hessian eigenanalyses to class separability. Importantly, the results also imply that the highest LDA ratio is observed for the combination of the first Hessian eigenvector with the first covariance eigenvector. This observation underscores the significance of projecting data onto the combined space of these primary eigenvectors (as outlined in Eqs (1, 2)), forming the core of our proposed method. These consistent empirical results across multiple datasets not only validate our theoretical premises but also endorse the effectiveness of our proposed method in optimizing class separability.

The evaluation results in Figure 2 compare the performance of different data projection methods, including PCA, KPCA, Hessian, UMAP, LLE, LOL, LDA, KDA, and the proposed method, using 5- or 10-fold cross-validation. It is essential to note that we introduced nonlinearity to the comparative analysis by utilizing distinct kernels determined through grid search in KPCA and KDA for each dataset. Notably, our proposed method consistently outperforms all others, securing the highest scores across all datasets and evaluation metrics. This consistent superiority of our approach implies its potential as a valuable tool for improving classification performance in various domains, further highlighting its promise as a robust and effective method for dimensionality reduction and data projection.

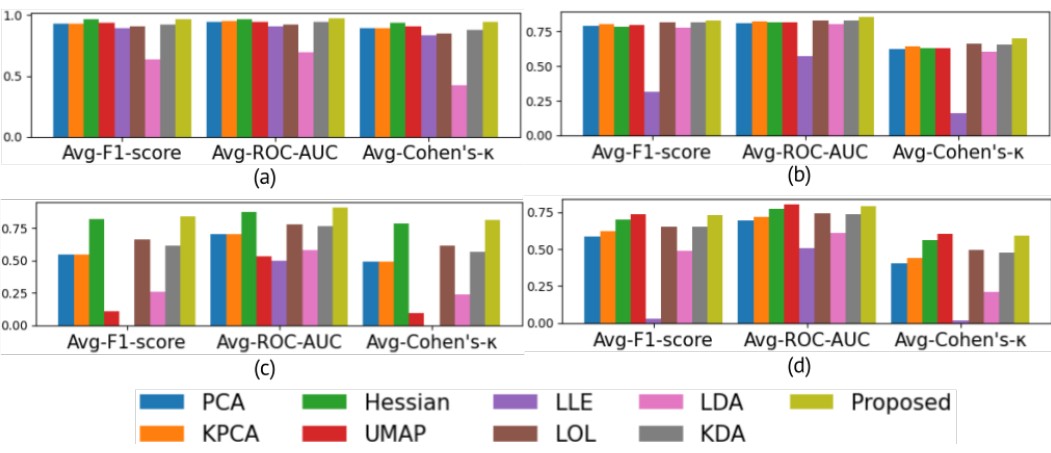

Figure 2: **Performance comparison of data projection methods using cross-validation on four distinct datasets: (a) WBCD, (b) heart disease, (c) neural spike train, and (d) Pima Indians diabetes datasets.** This figure presents the average F1 score, ROC AUC, and Cohen's Kappa values obtained through 5- or 10-fold cross-validation for nine data projection techniques: PCA, KPCA (cosine similarity for WBCD, polynomial kernel with degree=3 for Heart, linear kernel for Neural, cosine similarity for Pima), Hessian, UMAP, LLE, LOL, LDA, KDA (cosine similarity for WBCD, RBF kernel with coefficient=0.01 for Heart, sigmoid kernel with coefficient=2 for Neural, cosine similarity for Pima), and the proposed method. Notably, the proposed method consistently outperforms all other techniques, achieving the highest scores across all evaluation metrics.

## 5 DISCUSSION

Our work provides a compelling theoretical insight and a powerful, practical method, demonstrating the strength of simplicity in achieving remarkable results. Firstly, we provide a profound theoretical insight, revealing a subtle yet powerful relationship between covariance and Hessian matrices. Our formal proof seamlessly links covariance eigenanalysis with the first LDA criterion while Hessian eigenanalysis with the second one. This unification under LDA criteria offers a fresh and intuitive perspective on their interplay. Secondly, capitalizing on this theoretical elegance and simplicity, we introduce a novel method that consistently outperforms established techniques across diverse datasets. The unexpected efficacy of our method, rooted in the straightforward relationship between covariance, Hessian, and LDA, showcases the effectiveness of simplicity in addressing complex challenges.

The proposed method outperforms PCA and the Hessian method by comprehensively addressing both LDA criteria—maximizing between-class mean distance and minimizing within-class variances. Unlike PCA, which predominantly focuses on the former and lacks the guidance of class labels, our supervised approach considers both aspects. Despite the computational efficiency associated with unsupervised dimension reduction methods (Shen et al., 2014), our approach demonstrates the added value of incorporating class labels. This key insight also underlies our outperformance of KPCA, which, despite operating on non-linearities, remains essentially unsupervised in nature. While the Hessian method concentrates on minimizing within-class variances, our method optimally combines the strengths of PCA and the Hessian, effectively identifying feature space directions that enhance both between-class separation and within-class compactness.

While the proposed method employs the LDA criteria, it surpasses LDA itself in all cases. Figure 3 visually demonstrates the advantages of the proposed method over LDA. LDA is limited to a one-dimensional projection for binary classification problems Ye et al. (2004), where it seeks to identify a single direction that simultaneously satisfies the two criteria. Conversely, the proposed method splits the task of meeting the criteria into two directions. The utilization of higher dimensionality in the proposed method increases the likelihood of discovering class separability, aligning with Cover's theorem (Cover, 1965). KDA, operating on a non-linear mode, outperforms LDA in all cases, yet it remains fundamentally confined to one dimension, restricting its effectiveness in capturing intricate class-specific patterns compared to our proposed method.

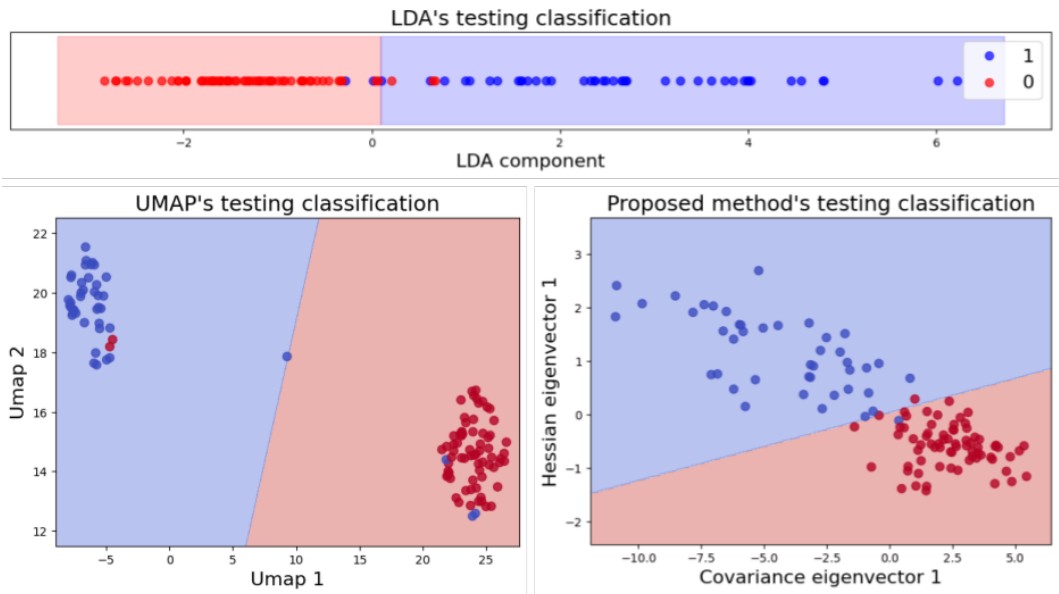

Figure 3: **Projection and classification results of UMAP, LDA, and the proposed method with SVM on the Wisconsin breast cancer test data.** This figure showcases the test data projected by three distinct data projection methods, each separated by its corresponding SVM hyperplane trained using the WBCD training data. This figure highlights the effectiveness of linear SVM in facilitating class separability visualization and interpretability of the model.

Our method outperforms LOL by leveraging non-linear modes of operation, providing a distinct advantage in capturing complex patterns beyond the linear capabilities of LOL. UMAP exhibits good class separability with the widest margin as shown in Figure 3. However, it is important to note that UMAP is not inherently a classification technique, and thus, it fails to generalize well to new data. Similar limitations apply to LLE, which, although effective in revealing local data structures, lacks the inherent capability for classification and generalization.

Figure 3 also underscores the simplicity and interpretability of linear SVMs as basic linear classifiers in dealing with low-dimensional data. The figure shows clear visualizations of the SVM's decision

boundaries and separation achieved through the various dimensionality reduction methods. Notably, our proposed method combined with linear SVMs offers valuable insights into the decision-making process within the underlying DNN. Leveraging the transparency and interpretability provided by SVMs to address the inherent opacity of DNNs, our method bridges the gap between complex DNNs and comprehensible, highly accurate decision processes within a 2D space.

The proposed method shows promising performance, but it has certain limitations. The applicability of our approach is based on the premise that a DNN model is already in place. Our method operates on top of the underlying DNN. Any shortcomings or biases in the DNN's performance will naturally reflect in the results obtained from our approach. The limitation of our method lies in its dependence on the quality and accuracy of the underlying DNN.

An important trajectory for future work involves investigating the extension of our methodology to accommodate various loss functions beyond binary cross-entropy. The mathematical derivation in our current work relies on the elegant relationship between binary cross-entropy loss and within-class variances. Exploring the adaptability of our method to different loss functions will contribute to a more comprehensive understanding of the method's versatility, but requires careful scrutiny to establish analogous connections. Simultaneously, we recognize the need to extend our methodology from binary to multiclass classification. The binary classification focus in this work stems from foundational aspects guiding our formal proof, which is designed around binary assumptions to facilitate a streamlined and elegant derivation process. In particular, the use of binary cross-entropy as the loss function and the utilization of a linear SVM for evaluation inherently adhere to binary classification. Moving forward, careful exploration is needed to adapt our approach to multiclass scenarios to ensure its applicability and effectiveness across a broader range of classification tasks.

## 6 CONCLUSION

Our paper presents a multifaceted contribution to the realms of binary classification and dimensionality reduction. We offer a rigorous formal proof, showcasing how our novel method, grounded in the eigenanalysis of covariance and Hessian matrices, systematically enhances class separability, drawing inspiration from LDA criteria. This theoretical foundation finds strong empirical support through comprehensive experiments across an array of datasets. In these experiments, our approach consistently outperforms established techniques, demonstrating its robustness and applicability. Moreover, our approach shares a similar complexity profile with the traditional method, ensuring its practical utility in real-world scenarios. Additionally, through the integration of our method with linear SVMs, we improve the explainability of the intricate decision-making processes inherent to DNNs, addressing their typical lack of transparency and facilitating enhanced model interpretability.

## 7 REPRODUCIBILITY STATEMENT

The detailed proofs for the theoretical foundations, emphasizing the maximization of between-class mean distance and minimization of within-class variance, are provided in Appendix A. The complete source code for our experiments is accessible through an anonymous Google account at the following links:

1. WBCD dataset experiment: Link to Colab notebook

2. Heart disease dataset experiment: Link to Colab notebook

3. Neural spike train dataset experiment: Link to Colab notebook

4. Pima Indians diabetes dataset experiment: Link to Colab notebook

These notebooks contain the complete source code, facilitating easy reproduction and comprehension of our results. Additionally, the datasets used in the experiments are available from the same Google Drive account, further enhancing the accessibility and reproducibility of our research findings.

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

# A FULL PROOFS: MAXIMIZING THE SQUARED BETWEEN-CLASS MEAN DISTANCE AND MINIMIZING THE WITHIN-CLASS VARIANCE

## A.1 PROOF OF THEOREM 1 - MAXIMIZING COVARIANCE FOR MAXIMIZING SQUARED BETWEEN-CLASS MEAN DISTANCE

To prove that maximizing the variance will maximize the squared between-class mean distance, we start by considering two sets of 1D data points representing two classes, denoted as $C_1$ and $C_2$, each consisting of $n$ samples. The data in $C_1$ and $C_2$ follow the same underlying distribution centered around their respective means, $m_1$ and $m_2$. In other words, $C_2$ can be understood as a reflection of $C_1$ with the axis of reflection positioned at the overall mean, denoted as $m$, effectively giving rise to a shifted configuration. Specifically, the indices for $C_1$ range from 1 to $n$, while the indices for $C_2$ range from $n + 1$ to $2n$.

For any pair of data points $x_i$ and $x_j$, where $1 \leq i \leq n$ and $n + 1 \leq j \leq 2n$, the difference between $x_i$ and $m_1$ is equal to the difference between $x_j$ and $m_2$, i.e., $x_i - m_1 = x_j - m_2$. This relationship ensures that the shifted copies of $C_1$ and $C_2$ maintain the same relative distances from their respective means, preserving the identical distribution in both sets. Furthermore, the variances of $C_1$ and $C_2$, are equal, denoted as $s_1^2 = s_2^2 = s_w^2$, and the combined data from $C_1$ and $C_2$ has a variance of $s^2$.

The between-class mean distance, denoted as $d$, represents the separation between the means of $C_1$ and $C_2$. We can express the means as $m_1 = m - \frac{d}{2}$ and $m_2 = m + \frac{d}{2}$, where $m$ is effectively located in the middle, equidistant from both $m_1$ and $m_2$.

The variance of the combined data is given by:

$$s^2 = \frac{1}{2n - 1} \left( \sum_{i=1}^{n} (x_i - m)^2 + \sum_{i=n+1}^{2n} (x_i - m)^2 \right)$$

To simplify the expression, we substitute $m = m_1 + \frac{1}{2}d$ and $m = m_2 - \frac{1}{2}d$. This yields:

$$s^2 = \frac{1}{2n - 1} \left( \sum_{i=1}^{n} \left( x_i - \left( m_1 + \frac{1}{2}d \right) \right)^2 + \sum_{i=n+1}^{2n} \left( x_i - \left( m_2 - \frac{1}{2}d \right) \right)^2 \right)$$

$$= \frac{1}{2n - 1} \left( \sum_{i=1}^{n} \left( (x_i - m_1) - \frac{1}{2}d \right)^2 + \sum_{i=n+1}^{2n} \left( (x_i - m_2) + \frac{1}{2}d \right)^2 \right)$$

$$= \frac{1}{2n-1} \left( \sum_{i=1}^{n} \left( (x_i - m_1)^2 - (x_i - m_1) d + \frac{1}{4} d^2 \right) \right.$$

$$\left. + \sum_{i=n+1}^{2n} \left( (x_i - m_2)^2 + (x_i - m_2) d + \frac{1}{4} d^2 \right) \right)$$

$$= \frac{1}{2n-1} \left( \sum_{i=1}^{n} \left( (x_i - m_1)^2 + \frac{1}{4} d^2 \right) - \sum_{i=1}^{n} ((x_i - m_1) d) \right.$$

$$\left. + \sum_{i=n+1}^{2n} \left( (x_i - m_2)^2 + \frac{1}{4} d^2 \right) \right) + \sum_{i=n+1}^{2n} ((x_i - m_2) d)$$

$$= \frac{1}{2n-1} \left( \sum_{i=1}^{n} \left( (x_i - m_1)^2 + \frac{1}{4} d^2 \right) + \sum_{i=n+1}^{2n} \left( (x_i - m_2)^2 + \frac{1}{4} d^2 \right) \right)$$

$$+ \sum_{i=n+1}^{2n} ((x_i - m_2) d) - \sum_{i=1}^{n} ((x_i - m_1) d)$$

$$= \frac{1}{2n-1} \left( \sum_{i=1}^{n} \left( (x_i - m_1)^2 + \frac{1}{4} d^2 \right) + \sum_{i=n+1}^{2n} \left( (x_i - m_2)^2 + \frac{1}{4} d^2 \right) \right)$$

$$+ \left( \sum_{i=n+1}^{2n} (x_i - m_2) - \sum_{i=1}^{n} (x_i - m_1) \right) d$$

The property of the identical distribution in both sets suggests $\sum_{i=n+1}^{2n} (x_i - m_2) - \sum_{i=1}^{n} (x_i - m_1) = 0$. So, we get:

$$s^2 = \frac{1}{2n-1} \left( \sum_{i=1}^{n} \left( (x_i - m_1)^2 + \frac{1}{4} d^2 \right) + \sum_{i=n+1}^{2n} \left( (x_i - m_2)^2 + \frac{1}{4} d^2 \right) \right)$$

$$= \frac{1}{2n-1} \left( \sum_{i=1}^{n} (x_i - m_1)^2 + \sum_{i=n+1}^{2n} (x_i - m_2)^2 + \frac{1}{2} n \cdot d^2 \right)$$

$$\approx \frac{1}{2(n-1)} \left( \sum_{i=1}^{n} (x_i - m_1)^2 + \sum_{i=n+1}^{2n} (x_i - m_2)^2 \right) + \frac{1}{4} d^2$$

$$= \frac{1}{2} \left( \frac{\sum_{i=1}^{n} (x_i - m_1)^2}{(n-1)} + \frac{\sum_{i=n+1}^{2n} (x_i - m_2)^2}{(n-1)} \right) + \frac{1}{4} d^2$$

$$= \frac{1}{2} \left( s_w^2 + s_w^2 \right) + \frac{1}{4} d^2$$

$$= s_w^2 + \frac{1}{4} d^2.$$

Now, considering the data points representing the projected data onto an (Eigen)vector, we can utilize the Variance Ratio Preservation Theorem for Projection onto a Vector, which establishes the relationship between $s^2$ and $s_w^2$ as follows:

$$s_w^2 = \lambda \cdot s^2$$

where $\lambda$ is a constant between 0 and 1, determined by the distribution of the original data being projected.

Substituting this equation into the previous expression, we have:

$$s^2 = \lambda \cdot s^2 + \frac{1}{4}d^2.$$

Let's rearrange the equation by moving $2\lambda \cdot s^2$ to the left side:

$$s^2 - \lambda \cdot s^2 = \frac{1}{4}d^2.$$

Combining like terms:

$$(1 - \lambda) \cdot s^2 = \frac{1}{4}d^2.$$

To solve for $s^2$, divide both sides by $(1 - \lambda)$:

$$s^2 = \frac{\frac{1}{4}d^2}{1 - \lambda} = \frac{r^2}{1 - \lambda}$$

where $r = \frac{1}{2}d = m - m_1 = m_2 - m$.

We observe that the sign of $s^2$ and $d^2$ will be the same since the denominator $1 - \lambda$ is always positive (as $0 < \lambda < 1$). Therefore, $s^2$ is linearly proportional to $d^2$.

Hence, maximizing the variance ($s^2$) will maximize the squared between-class mean distance ($d^2$) as desired.

## A.2 PROOF OF THEOREM 2 - MAXIMIZING HESSIAN FOR MINIMIZING WITHIN-CLASS VARIANCE

We aim to prove that maximizing the Hessian will minimize the within-class variance. Let $\theta$ denote a parameter of the classifier.

We define the within-class variance as the variance of a posterior distribution $p_\theta(\theta \mid c_i)$, which represents the distribution of the parameter $\theta$ given a class $c_i$. We denote the variance of this posterior distribution as $\sigma^2_{post}$.

Recall that our Hessian is given by:

$$\mathrm{H}_\theta = -\left[\nabla^2_\theta \log p_\theta(c_i \mid \theta)\right]$$

However, according to Barshan et al. (2020), we can approximate the Hessian using Fisher information:

$$\mathrm{H}_\theta \approx \mathbb{E}_{p_\theta}\left[\left(\nabla_\theta \log p_\theta(c_i \mid \theta)\right)^2\right]$$

Assuming that a known normal distribution underlies the likelihood $p_\theta(c_i \mid \theta)$, i.e.,

$$p_\theta(c_i \mid \theta) = \frac{1}{\sqrt{2\pi\sigma^2}}\exp\left(-\frac{(\theta - \mu)^2}{2\sigma^2}\right)$$

where $\mu$ and $\sigma$ are known constants, we can compute the Hessian as follows:

$$H_\theta = \mathbb{E}_{p_\theta} \left[ \left( \nabla_\theta \log \frac{1}{\sqrt{2\pi\sigma^2}} \exp\left( -\frac{(\theta-\mu)^2}{2\sigma^2} \right) \right)^2 \right]$$

$$= \mathbb{E}_{p_\theta} \left[ \left( \frac{\theta-\mu}{\sigma^2} \right)^2 \right]$$

$$= \frac{1}{\sigma^4} \mathbb{E}_{p_\theta} \left[ (\theta-\mu)^2 \right]$$

$$= \frac{1}{\sigma^4} \sigma^2$$

$$= \frac{1}{\sigma^2}$$

Now, let us assume that the evidence $p(c_i)$ is a known constant $\rho$. We also assume that the prior distribution $p(\theta)$ follows a uniform distribution within the plausible range of $\theta$, which is bounded by a known minimum value $\theta_{\min}$ and maximum value $\theta_{\max}$. Formally:

$$p_\theta(\theta) = \begin{cases} \dfrac{1}{\theta_{\max} - \theta_{\min}}, & \text{if } \theta_{\min} \le \theta \le \theta_{\max} \\ 0, & \text{otherwise} \end{cases}$$

Based on Bayes' formula and the given assumptions, the posterior distribution within the plausible range of $\theta$ is:

$$p_\theta(\theta \mid c_i) = \frac{p_\theta(c_i \mid \theta) \cdot p_\theta(\theta)}{p(c_i)}$$

$$= \frac{1}{\rho(\theta_{\max} - \theta_{\min})} \cdot \frac{1}{\sqrt{2\pi\sigma^2}} \exp\left( -\frac{(\theta-\mu)^2}{2\sigma^2} \right)$$

This implies that the posterior distribution $p_\theta(\theta \mid c_i)$ is a normal distribution with mean $\mu$ and variance $\sigma^2$:

$$p_\theta(\theta \mid c_i) \sim \mathcal{N}(\mu, \sigma^2)$$

Therefore, the within-class variance $\sigma^2_{\text{post}}$ of the posterior distribution is equal to $\sigma^2$.

Combining the previous result with the Hessian calculation, we can conclude that:

$$H_\theta = \frac{1}{\sigma^2} = \frac{1}{\sigma^2_{\text{post}}}$$

Hence, maximizing the Hessian matrix ($H_\theta$) will minimize the within-class variances ($\sigma^2_{\text{post}}$) as desired.

## A.3 SUPPORTING THEOREMS

### A.3.1 DISTANCE PRESERVATION THEOREM FOR PROJECTION ONTO A VECTOR

In a 2-dimensional space with a first axis and a second axis, consider a set of points located solely on the first axis, denoted by $(x_i, 0)$, where $x_i$ represents the position of a point along the first axis. Let $\mathbf{v} = (v_1, v_2)$ be a unit vector, with $v_1$ and $v_2$ as its components. The projection operation maps each point $(x_i, 0)$ to its corresponding scalar projection $y_i$ onto the vector $\mathbf{v}$. The Distance Preservation Theorem states that the projection onto $\mathbf{v}$ preserves the ratio of distances between the points on the first axis.

**Proof:** The scalar projection $y_i$ of a point $(x_i, 0)$ onto the vector $\mathbf{v}$ is given by:

$$y_i = \frac{(x_i, 0) \cdot \mathbf{v}}{|\mathbf{v}|} = \frac{(x_i, 0) \cdot \mathbf{v}}{1} = x_i \cdot v_1 + 0 \cdot v_2 = x_i \cdot v_1$$

This implies that the projection of each point $(x_i, 0)$ from the first axis onto the vector $\mathbf{v}$ is obtained by multiplying the coordinate $x_i$ by the first component $v_1$ of $\mathbf{v}$.

Now, considering the distances between two points $(x_i, 0)$ and $(x_j, 0)$ on the first axis and their corresponding projections $y_i$ and $y_j$ onto the vector $\mathbf{v}$, we define the distance between $(x_i, 0)$ and $(x_j, 0)$ as:

$$d_1 = |x_i - x_j|$$

Similarly, the distance between $y_i$ and $y_j$ is defined as:

$$d_2 = |y_i - y_j|$$

Substituting the expressions for $y_i$ and $y_j$ derived earlier, we have:

$$d_2 = |x_i \cdot v_1 - x_j \cdot v_1| = |v_1| \cdot |x_i - x_j|$$

It can be observed that the ratio of the distances is constant:

$$\frac{d_2}{d_1} = \frac{|v_1| \cdot |x_i - x_j|}{|x_i - x_j|} = |v_1|$$

This shows that the ratio of distances between the points on the first axis is preserved in the projection onto $\mathbf{v}$.

Therefore, the Distance Preservation Theorem for Projection onto a Vector concludes that the projection operation onto the vector $\mathbf{v} = (v_1, v_2)$ in a 2-dimensional space preserves the ratio of distances between the points on the first axis.

### A.3.2 VARIANCE PRESERVATION THEOREM FOR PROJECTION ONTO A VECTOR

Consider an arbitrary subset of points located on the first axis, denoted by $X = \{x_1, x_2, \ldots, x_n\}$, with a mean $\mu_X$ and variance $\sigma_X^2$. These points are projected onto the vector $\mathbf{v} = (v_1, v_2)$ using the projection operation defined earlier.

The projected subset of points on $\mathbf{v}$ is denoted by $Y = \{y_1, y_2, \ldots, y_n\}$, where each $y_i$ represents the projection of $x_i$ onto $\mathbf{v}$. The variance of $Y$, denoted as $\sigma_Y^2$, is a measure of the spread of the projected points around its mean $\mu_Y$.

The Variance Preservation Theorem states that the ratio of variances between the projected subset $Y$ and the original subset $X$ is equal to the square of the first component $v_1$ of the projection vector $\mathbf{v}$.

Mathematically, this can be expressed as:

$$\frac{\sigma_Y^2}{\sigma_X^2} = v_1^2$$

**Proof:** The variance of $Y$ can be computed as:

$$\sigma_Y^2 = \frac{1}{n-1} \sum_{i=1}^{n} (y_i - \mu_Y)^2$$

Similarly, the variance of $X$ can be computed as:

$$\sigma_X^2 = \frac{1}{n-1} \sum_{i=1}^{n} (x_i - \mu_X)^2$$

According to the Distance Preservation Theorem for Projection onto a Vector, the squared distances between the points on the first axis and their projections can be related as:

$$(y_i - \mu_Y)^2 = |y_i - \mu_Y|^2 = (|v_1| \cdot |x_i - \mu_X|)^2 = v_1^2 \cdot (x_i - \mu_X)^2$$

Substituting this expression into the variance of $Y$, we have:

$$\sigma_Y^2 = \frac{1}{n-1} \sum_{i=1}^{n} v_1^2 \cdot (x_i - \mu_X)^2 = v_1^2 \cdot \frac{1}{n-1} \sum_{i=1}^{n} (x_i - \mu_X)^2 = v_1^2 \cdot \sigma_X^2$$

Therefore, we have shown that:

$$\sigma_Y^2 = v_1^2 \cdot \sigma_X^2$$

Hence, the ratio of variances between the projected subset $Y$ and the original subset $X$ is given by $v_1^2$, as stated in the Variance Preservation Theorem for Projection onto a Vector.

### A.3.3 VARIANCE RATIO PRESERVATION THEOREM FOR PROJECTION ONTO A VECTOR

Consider an arbitrary subset of points located on the first axis, denoted by $X = \{x_1, x_2, \ldots, x_n\}$, with variances $\sigma_{X_1}^2$ and $\sigma_{X_2}^2$ for subsets $X_1$ and $X_2$, respectively. These points are projected onto the vector $\mathbf{v} = (v_1, v_2)$ using the projection operation defined earlier.

The projected subsets of points on $\mathbf{v}$ are denoted by $Y_1 = \{y_1, y_2, \ldots, y_n\}$ and $Y_2 = \{z_1, z_2, \ldots, z_n\}$, where each $y_i$ and $z_i$ represents the projection of $x_i$ onto $\mathbf{v}$. The variances of $Y_1$ and $Y_2$, denoted as $\sigma_{Y_1}^2$ and $\sigma_{Y_2}^2$ respectively, measure the spread of the projected points around their respective means.

The Variance Ratio Preservation Theorem states that the ratio of variances between the projected subsets $Y_2$ and $Y_1$ is equal to the ratio of variances between the original subsets $X_2$ and $X_1$.

Mathematically, this can be expressed as:

$$\frac{\sigma_{Y_2}^2}{\sigma_{Y_1}^2} = \frac{\sigma_{X_2}^2}{\sigma_{X_1}^2}$$

**Proof:** By the Variance Preservation Theorem for Projection onto a Vector, we know that the variance of the projected subset $Y$ can be expressed as:

$$\sigma_Y^2 = v_1^2 \cdot \sigma_X^2$$

Applying this theorem to $Y_1$ and $X_1$, we have:

$$\sigma_{Y_1}^2 = v_1^2 \cdot \sigma_{X_1}^2$$

Similarly, applying the theorem to $Y_2$ and $X_2$, we have:

$$\sigma_{Y_2}^2 = v_1^2 \cdot \sigma_{X_2}^2$$

Now, we can take the ratio of these variances:

$$\frac{\sigma_{Y_2}^2}{\sigma_{Y_1}^2} = \frac{v_1^2 \cdot \sigma_{X_2}^2}{v_1^2 \cdot \sigma_{X_1}^2} = \frac{\sigma_{X_2}^2}{\sigma_{X_1}^2}$$

Therefore, we have shown that the ratio of variances between the projected subsets $Y_2$ and $Y_1$ is equal to the ratio of variances between the original subsets $X_2$ and $X_1$, as required.

# B   ADDITIONAL DATASET RESULTS

Figure 4: **Projection of the heart disease data into different combined spaces of the covariance and Hessian eigenvectors.**

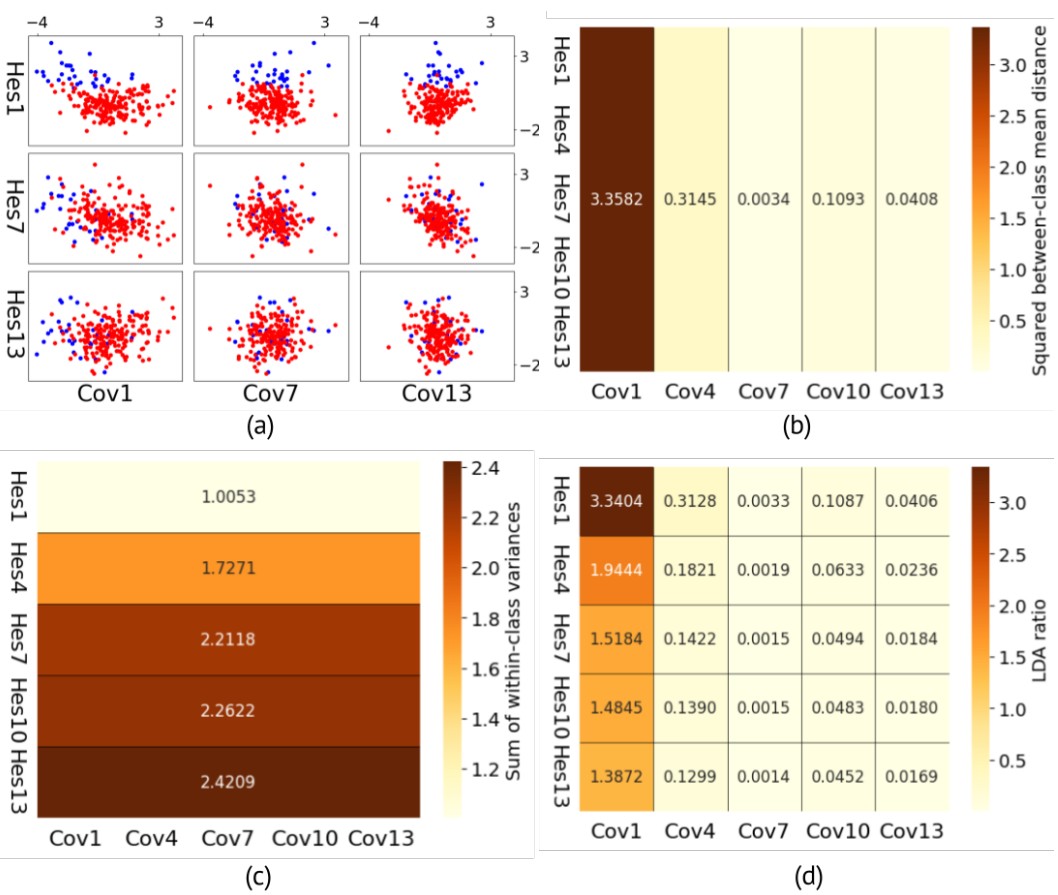

Figure 5: **Projection of the neural spike train data into different combined spaces of the covariance and Hessian eigenvectors.**

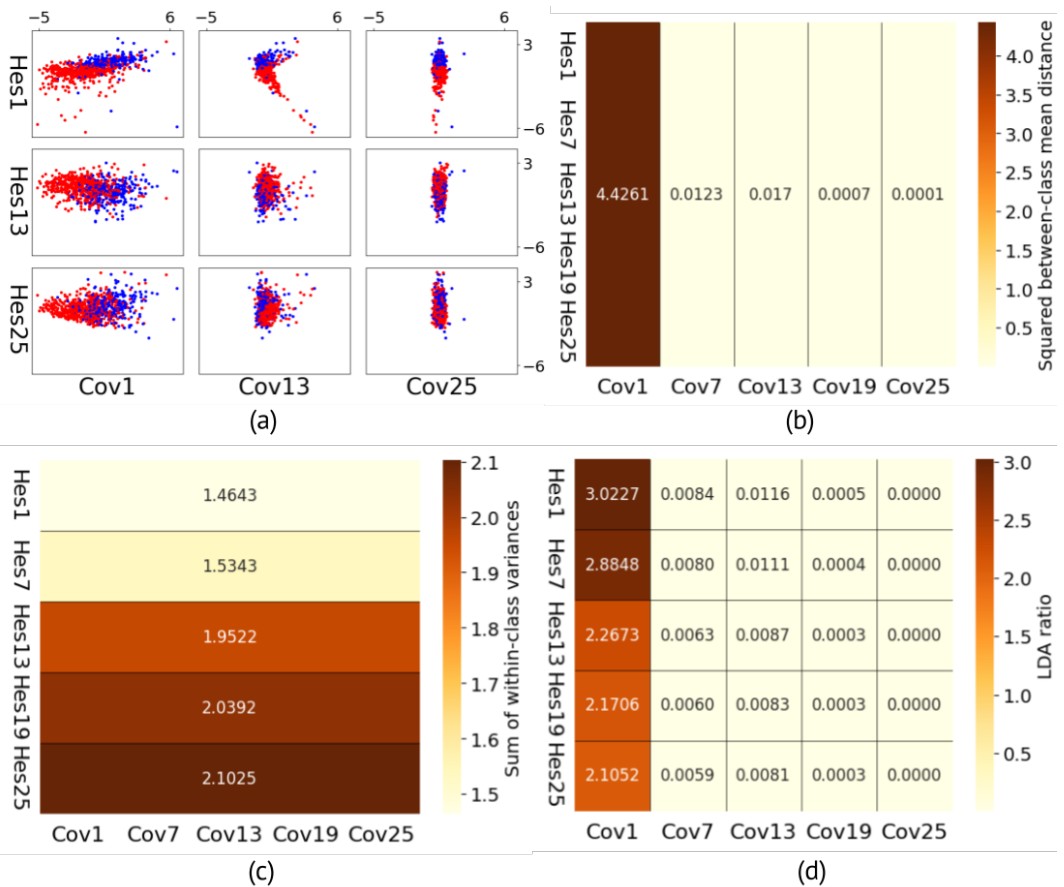

Figure 6: **Projection of the Pima Indians diabetes data into different combined spaces of the covariance and Hessian eigenvectors.** **(a)** Nine selected projection plots, each representing data projected onto a distinct space created by combining the first three covariance and first three Hessian eigenvectors. **(b)** Heatmap showing the squared between-class mean distance for projections onto varying combinations of covariance and Hessian eigenvectors. **(c)** Heatmap showing the sum of within-class variances for projections onto different combinations of covariance and Hessian eigenvectors. **(d)** Heatmap displaying the LDA ratio, representing the ratio between the squared between-class mean distances presented in (b) and the corresponding within-class variances shown in (c).

