# OpenReview forum: "Exploring the Combined Power of Covariance and Hessian Matrices Eigenanalysis for Binary Classification"
_ICLR.cc/2024/Conference — Submitted to ICLR 2024_

### Official Review · Reviewer_YZvz · 2023-10-28

**Soundness:** 2 fair
**Presentation:** 3 good
**Contribution:** 2 fair
**Rating:** 5
**Confidence:** 2

**Summary:**

This paper proposes a data projection approach which combines the power of covariance and Hessian matrices in the binary classification task. Specifically, the method combines eigenanalysis of a covariance matrix evaluated on a training set with a Hessian matrix evaluated on a deep learning model to achieve optimal class separability. Benefiting from the linear discriminant analysis (LDA) criteria, the proposed method achieves better class separability in contrast to PCA and the Hessian method. Empirical results show its better performance in binary classification.

**Strengths:**

1)Combining the power of the covariance matrix and the Hessian matrix is interesting and novel, to my knowledge. And this work gives a new learning perspective for binary classification problems.

2)The writing style is good and the motivation is clear.

**Weaknesses:**

1)The description of the key technique is unclear. How to integrate the covariance matrix and the Hessian matrix can be confusing. The authors only offer some description in the Section 2.1 and it lacks more detailed theoretical explanation.

2)The comparison methods are insufficient. To clarify the superiority of the proposed method, the authors compare the method with four data projection techniques including PCA, Hessian, UMAP and LDA. However, some other dimensionality reduction and data projection techniques should be included. For example, kernel based methods including kernel PCA and kernel LDA and manifold based methods like locally linear embedding (LLE) and t-distributed Stochastic Neighbor Embedding (t-SNE) are also representative methods in this problem.

3)The classification results in Figure 2 can be confusing. The performance of the proposed method is not obviously superior than other competing methods at times. For example, the performance of the proposed method is very close to the performance of Hessian in the WBCD database. Moreover, the performance of the proposed method is very close to the performance of UMAP in the Pima Indians diabetes database.

**Questions:**

1)As mentioned above, the authors should offer more details of the key technique about how to integrate the covariance matrix and the Hessian matrix. For better  readability, an interpretative figure to illustrate the key technique is needed.

2)The authors should provide a specific algorithm of the proposed method. The procedure of the algorithm remains highly unclear.

3)The authors should offer more experimental results to validate the effectiveness of the proposed method, which are not limited to more comparison methods and more convincing classification results. As mentioned above, some other dimensionality reduction and data projection techniques should be included. Besides, the performance on the classification problem is not outstanding,

4) This work is limited in the problem of binary classification. And it would be better if the authors could devise a multi-class classifier considering the real-world applications.

---

> ### Author Response · Authors · 2023-11-17
> **Response to Reviewer YZvz (part 1)**
>
> Dear Reviewer YZvz,
>
> We appreciate the time and effort you invested in reviewing our manuscript. We are pleased that you acknowledge the interesting and novel nature of our proposed method, and found the writing style clear and the motivation well-presented. Moreover, your insights have been invaluable in refining the quality of our work.
>
> We would like to inform you that we have already submitted a revised version of the manuscript that addresses the comments and concerns you raised.
>
> Below are our responses to your specific points:
>
> 1. **Response to question 1 and 2:**
>    - > "As mentioned above, the authors should offer more details of the key technique about how to integrate the covariance matrix and the Hessian matrix. For better readability, an interpretative figure to illustrate the key technique is needed."
>    - > "The authors should provide a specific algorithm of the proposed method. The procedure of the algorithm remains highly unclear."
>
> We acknowledge the need for a more explicit presentation of the algorithm for the proposed method. In response to this, we have added additional clarity and detail in Section 2.1 of the revised manuscript, providing a more thorough description of the key technique.
>
> In the revised Section 2.1:
>
> 1. **Covariance matrix eigenanalysis:**
>    We clarified the eigenanalysis of the covariance matrix ($Cov(\boldsymbol{\theta})$) and its role in capturing the principal directions with the highest variances. The leading eigenvector ($\mathbf{v}_1$) associated with the largest eigenvalue ($\lambda_1$) represents the principal direction with the highest variance.
>
>    The eigen-equation is given by $Cov(\boldsymbol{\theta}) \cdot \mathbf{v}_1 = \lambda_1 \cdot \mathbf{v}_1$.
>
> 2. **Hessian matrix eigenanalysis:**
>    We explained the eigenanalysis of the Hessian matrix ($H_{\boldsymbol{\theta}}$) in detail, including the utilization of a deep neural network with binary cross-entropy loss during training. The leading eigenvector ($\mathbf{v}_1'$) associated with the largest eigenvalue ($\lambda_1'$) represents the direction corresponding to the sharpest curvature.
>
>
>    The eigen-equation is given by $H_{\boldsymbol{\theta}} \cdot \mathbf{v}_1' = \lambda_1' \cdot \mathbf{v}_1'$.
>
> 3. **Integration of matrices and projection of data:**
>    We explicitly presented the integration of matrices and projection of data into a 2D space with mathematical formulations.
>
>    $$ \mathbf{U} = [\mathbf{v}_1, \mathbf{v}_1'] $$
>
>    The 2D projection of the data is obtained by:
>
>    $$ \mathbf{X}_{\text{proj}} = \mathbf{X} \cdot \mathbf{U} $$
>
>    where $\mathbf{X}$ is the original data matrix, and $\mathbf{X}_{\text{proj}}$ represents the final output of the proposed method.
>
> We want to emphasize that this summary provides the core mathematical ideas, and the complete and detailed explanations, along with additional context, can be found in the revised manuscript.

---

> ### Author Response · Authors · 2023-11-17
> **Response to Reviewer YZvz (part 2)**
>
> 2. **Response to question 3:**
>    - > "The comparison methods are insufficient. To clarify the superiority of the proposed method, the authors compare the method with four data projection techniques including PCA, Hessian, UMAP and LDA. However, some other dimensionality reduction and data projection techniques should be included. For example, kernel based methods including kernel PCA and kernel LDA and manifold based methods like locally linear embedding (LLE) and t-distributed Stochastic Neighbor Embedding (t-SNE) are also representative methods in this problem."
>    - > "The authors should offer more experimental results to validate the effectiveness of the proposed method, which are not limited to more comparison methods and more convincing classification results. As mentioned above, some other dimensionality reduction and data projection techniques should be included. Besides, the performance on the classification problem is not outstanding."
>
> -	To address the concern regarding the comparison methods, we have expanded our evaluation to include a broader spectrum of contemporary techniques. In the revised manuscript, we compare our proposed method with nine distinct approaches, encompassing kernel-based methods like KPCA and KDA, manifold based methods like UMAP and LLE as well as LOL [1]. This extended comparison aims to provide a more comprehensive view of the proposed method's performance.
> -	We have introduced nonlinearity through distinct kernels determined by grid search in KPCA and KDA for each dataset.
> -	The results, discussed in the revised manuscript, demonstrate the consistent superiority of our approach across various datasets and metrics.
> -	These extended results are completely reproducible using the same Colab notebooks whose links are provided for transparency and validation.
> -	The relevant discussion for these additional comparisons has been appropriately incorporated into the Discussion section of the revised manuscript.
> -	The abstract has also been updated to reflect the implications of these extended comparisons.
>
>
> 3. **Response to question 4:**
>    - > "This work is limited in the problem of binary classification. And it would be better if the authors could devise a multi-class classifier considering the real-world applications."
>
> While our work focuses on binary classification, we appreciate your suggestion to extend our methodology to multiclass classification as a potential avenue for future research. We have explicitly mentioned in the Discussion section of the revised manuscript that the binary classification focus in this work stems from foundational aspects guiding our formal proof, which is designed around binary assumptions to facilitate a streamlined and elegant derivation process. In particular, the use of binary cross-entropy as the loss function and the utilization of a linear SVM for evaluation inherently adhere to binary classification. Moving forward, careful exploration is needed to adapt our approach to multiclass scenarios to ensure its applicability and effectiveness across a broader range of classification tasks.
>
> We hope these revisions address your concerns, and we welcome any further feedback or suggestions you may have. Thank you once again for your thorough review and valuable input.
>
> Reference:
>
> [1] Vogelstein, J. T., Bridgeford, E. W., Tang, M., Zheng, D., Douville, C., Burns, R., & Maggioni, M. (2021). Supervised dimensionality reduction for big data. Nature communications, 12(1), 2872.

---

### Official Review · Reviewer_LZ1V · 2023-10-30

**Soundness:** 3 good
**Presentation:** 3 good
**Contribution:** 4 excellent
**Rating:** 6
**Confidence:** 3

**Summary:**

This paper proposes an efficient binary classification method based on integrating the covariance and Hessian matrices in improving classification performance. This method combines the eigenanalysis of a covariance matrix evaluated on a training set with a Hessian matrix evaluated on a deep learning model to achieve optimal class separability in binary classification tasks. Both theoretical proofs and experimental results are demonstrated to consolidate the theory.

**Strengths:**

proposes a method that combines covariance and Hessian matrices to perform classification analysis more effectively

**Weaknesses:**

The part 3 of Methodologies session (Section 2.1) can be written more clearly.

**Questions:**

I wonder if the part 3 of Methodologies session (Section 2.1) can be written more clearly? In particular, why the result of the claimed process yields a 2D projection of the data?

---

> ### Author Response · Authors · 2023-11-16
> **Response to Reviewer LZ1V**
>
> Dear Reviewer LZ1V,
>
> Thank you for your thoughtful review and valuable feedback on our manuscript, Submission7389. We appreciate the time and effort you invested in providing constructive comments.
>
> We would like to inform you that the revised and clarified content of Section 2.1 has been incorporated into the updated manuscript that we have submitted. We hope that the revisions address your concerns and enhance the clarity of the proposed approach.
>
> In the revised Section 2.1:
>
> 1. **Covariance matrix eigenanalysis:**
>    We clarified the eigenanalysis of the covariance matrix ($Cov(\boldsymbol{\theta})$) and its role in capturing the principal directions with the highest variances. The leading eigenvector ($\mathbf{v}_1$) associated with the largest eigenvalue ($\lambda_1$) represents the principal direction with the highest variance.
>
>    The eigen-equation is given by $Cov(\boldsymbol{\theta}) \cdot \mathbf{v}_1 = \lambda_1 \cdot \mathbf{v}_1$.
>
> 2. **Hessian matrix eigenanalysis:**
>    We explained the eigenanalysis of the Hessian matrix ($H_{\boldsymbol{\theta}}$) in detail, including the utilization of a deep neural network with binary cross-entropy loss during training. The leading eigenvector ($\mathbf{v}_1'$) associated with the largest eigenvalue ($\lambda_1'$) represents the direction corresponding to the sharpest curvature.
>
>
>    The eigen-equation is given by $H_{\boldsymbol{\theta}} \cdot \mathbf{v}_1' = \lambda_1' \cdot \mathbf{v}_1'$.
>
> 3. **Integration of matrices and projection of data:**
>    We explicitly presented the integration of matrices and projection of data into a 2D space with mathematical formulations.
>
>    $$ \mathbf{U} = [\mathbf{v}_1, \mathbf{v}_1'] $$
>
>    The 2D projection of the data is obtained by:
>
>    $$ \mathbf{X}_{\text{proj}} = \mathbf{X} \cdot \mathbf{U} $$
>
>    where $\mathbf{X}$ is the original data matrix, and $\mathbf{X}_{\text{proj}}$ represents the final output of the proposed method.
>
> We want to emphasize that this summary provides the core mathematical ideas, and the complete and detailed explanations, along with additional context, can be found in the revised manuscript.
>
> If you have any specific points or further inquiries, we are more than willing to address them. We look forward to hearing your thoughts on the revised manuscript.

---

> > ### Comment · Reviewer_LZ1V · 2023-11-23
> > **increase score to 4**
> >
> > Thanks for the response. I would like to increase my score to 4.

---

> > > ### Author Response · Authors · 2023-11-23
> > > **Acknowledging Reviewer LZ1V's positive score adjustment**
> > >
> > > Dear Reviewer LZ1V,
> > >
> > > Your feedback is invaluable to us, and we want to ensure that all your questions and expectations are addressed satisfactorily. We appreciate your consideration and would like to know if there are any additional concerns or information you may need from our end to facilitate the score adjustment. If you have no further requests, we look forward to the score increase.
> > >
> > > Thank you once again for your time and feedback.

---

### Official Review · Reviewer_E7ZV · 2023-11-06

**Soundness:** 2 fair
**Presentation:** 2 fair
**Contribution:** 2 fair
**Rating:** 6
**Confidence:** 4

**Summary:**

This manuscript theoretically and numerically evaluates the projection matrices derived from both the Covariance and the Hessian, in terms  of how they impact classification performance.

**Strengths:**

Covariance and Hessian matrices of various kinds are indeed of critical importance in classification performance, and warrant further study.

**Weaknesses:**

1. I found the theory to be a weak.  For me to believe the relevance of this theory, it must operate within a multivariate context.  The question is about whether the top eigenvalues of either of these matrices contain the relevant signal. Theory operating on unidimensional data I think has relatively little to offer on this topic.

2. As I read the paper, the authors talk about *the* Hessian matrix.  However, the paper is about a Hessian estimated using a specific deep net, which is of course *a* Hessian, but not *the* Hessian.  The discussion implied (to me) much more general claims than were warranted, imho, given the actual theoretical and empirical results. I would have expected a specific mention of Fisher's Information Matrix, which is closely related to the Hessian, as it includes it, and is a known bound of the variance for any random variable.

3. LDA is well known to find the projection that balances maximizing across-variance while minimizing within-variance.  It was never clear to me why we would want another method to do something like that?  What is missing in LDA that this method achieves? I can imagine a desire to embed in multiple dimensions, rather than just 1, but see my next point about that.

4.  Under the Gaussian model, the direction that captures the variance across classes is simply the difference of means vector (after 'whitening'), and the direction that maximizes the variance within is the class-centered covariance.  Reduced Rank LDA essentially combines those two: it projects the data onto the matrix which is the product of the difference of means with the low-rank estimate of the pooled covariance.  So, we already have a standard/classical approach to embedding into these two dimensions.  How is your approach better than this?

5. The defined Hessian is closely related to another standard thing called the Pointwise Mutual Information (https://en.wikipedia.org/wiki/Pointwise_mutual_information), which is very commonly used in language processing and embedding.  A discussion/comparison with this method would be desirable.

6. The numerical results indicate that embedding using the proposed approach is slightly better than the unsupervised approaches, or LDA, which is purely linear.  But your approach is nonlinear.  So, unless the data are strongly linear, a supervised nonlinear approach is likely to win.  And your proposed approach must lose in simulations where the data truly are linear. I'd think any reasonable kernel LDA approach would improve relative to PCA or LDA, assuming a large enough sample size.

**Questions:**

There are a few relevant papers that might be worthwhile reading for more background, including our paper, https://www.nature.com/articles/s41467-021-23102-2, and one ours built on, https://www.sciencedirect.com/science/article/pii/S0047259X14001201?via%3Dihub.

---

> ### Author Response · Authors · 2023-11-20
> **Response to Reviewer E7ZV (part 1)**
>
> Dear Reviewer E7ZV,
>
> Thank you for dedicating your time to review our manuscript. We sincerely appreciate your thoughtful comments.
>
> We want to inform you that we have already submitted a revised version of the manuscript that addresses the concerns and questions you raised. We have incorporated significant revisions into the manuscript, focusing on clarity, detailed explanations, and additional comparisons to enhance the overall quality of the paper. The updated version aims to provide a more comprehensive and satisfactory reading experience.
>
> Below is our detailed response to each of your points:
>
> 1. **Response to concern 1:**
>    - > "I found the theory to be a weak. For me to believe the relevance of this theory, it must operate within a multivariate context. The question is about whether the top eigenvalues of either of these matrices contain the relevant signal. Theory operating on unidimensional data I think has relatively little to offer on this topic."
>
> We understand your concern about the perceived weakness of the theory, particularly in a simple, univariate context. In response to this, we have added a paragraph at the beginning of the Discussion section in the revised manuscript. In our revised discussion, we emphasize the strength of simplicity in achieving remarkable results. In our work, we firstly provide a profound theoretical insight, revealing a subtle yet powerful relationship between covariance and Hessian matrices. Our formal proof seamlessly links covariance eigenanalysis with the first LDA criterion while Hessian eigenanalysis with the second one. This unification under LDA criteria offers a fresh and intuitive perspective on their interplay. Secondly, we highlight that simplicity, when harnessed effectively, can lead to powerful and practical solutions. Drawing inspiration from the elegant and simple theoretical relationship, we introduce a novel method that consistently outperforms established techniques across diverse datasets. This unexpected efficacy is rooted in the straightforward relationship between covariance, Hessian, and LDA, showcasing the effectiveness of simplicity in addressing complex challenges.
>
> Additionally, we find resonance in the work on Linear Optimal Low-rank projection. The methodology, despite its simplicity, has demonstrated remarkable success in enhancing data representations for various classification tasks while maintaining computational efficiency and scalability [1]. This further supports our belief in the potential of simplicity to achieve significant advancements in the field.
>
> 2. **Response to concern 2:**
>    - > " As I read the paper, the authors talk about the Hessian matrix. However, the paper is about a Hessian estimated using a specific deep net, which is of course a Hessian, but not the Hessian. The discussion implied (to me) much more general claims than were warranted, imho, given the actual theoretical and empirical results. I would have expected a specific mention of Fisher's Information Matrix, which is closely related to the Hessian, as it includes it, and is a known bound of the variance for any random variable."
>
> You rightly pointed out that in the original manuscript, there was a usage of "the Hessian matrix," which has been corrected in the revised version. We acknowledge this oversight, and the updated manuscript now consistently refers to "a Hessian matrix." We have also explicitly mentioned Fisher's information as an approximation of the Hessian in Section 2.2 of the revised manuscript. This modification ensures a clear acknowledgment of the connection between the Hessian matrix and Fisher's information matrix, providing a more accurate representation of our methodology.
>
>
> Reference:
>
> [1]	Vogelstein, J. T., Bridgeford, E. W., Tang, M., Zheng, D., Douville, C., Burns, R., & Maggioni, M. (2021). Supervised dimensionality reduction for big data. Nature communications, 12(1), 2872.

---

> > ### Comment · Reviewer_E7ZV · 2023-11-22
> > **concern 1 and 2**
> >
> > 1. I am not convinced by the paragraph.  Showing a theoretical connection between univariate and multivariate theory would be required for me to be satisfied.
> >
> > 2. Great!

---

> ### Author Response · Authors · 2023-11-20
> **Response to Reviewer E7ZV (part 2)**
>
> 3. **Response to concern 3 and 4:**
>    - > " LDA is well known to find the projection that balances maximizing across-variance while minimizing within-variance. It was never clear to me why we would want another method to do something like that? What is missing in LDA that this method achieves? I can imagine a desire to embed in multiple dimensions, rather than just 1, but see my next point about that."
>    - > " Under the Gaussian model, the direction that captures the variance across classes is simply the difference of means vector (after 'whitening'), and the direction that maximizes the variance within is the class-centered covariance. Reduced Rank LDA essentially combines those two: it projects the data onto the matrix which is the product of the difference of means with the low-rank estimate of the pooled covariance. So, we already have a standard/classical approach to embedding into these two dimensions. How is your approach better than this?"
>
> Our initial motivation was not driven by a desire to embed in multiple dimensions, as mentioned. Instead, it originated from the intention to explore the relationship between two familiar concepts: covariance and Hessian matrices. Unexpectedly, this exploration led to their unification under another familiar concept, namely the LDA concept. The elegance of this relationship, as highlighted in the revised manuscript, allows us to introduce a method that outperforms established methods, both linear and non-linear, including Linear DA and Kernel DA. The method's superiority is not based on a need for multiple dimensions but rather on the unique insights provided by this unification.
>
> We appreciate the opportunity to clarify why our approach outperforms LDA. Experimental results unequivocally demonstrate the superior performance of our method across all cases, supported by a robust theoretical explanation. Our method surpasses LDA by leveraging higher-dimensional feature spaces, aligning with Cover’s theorem, which favors linear separability in higher dimensions.
>
> 4. **Response to concern 5:**
>    - > " The defined Hessian is closely related to another standard thing called the Pointwise Mutual Information (https://en.wikipedia.org/wiki/Pointwise_mutual_information), which is very commonly used in language processing and embedding. A discussion/comparison with this method would be desirable."
>
> We appreciate your suggestion regarding the potential connection between the defined Hessian and Pointwise Mutual Information (PMI). However, after careful consideration, we believe that PMI might not be directly relevant to the goals and scope of our work, which focuses on binary classification tasks and the interplay between covariance and Hessian matrices. While PMI is indeed a valuable tool in language processing and embedding, the objectives and underlying principles of our work differ significantly.
>
> If you have further questions or if there are specific aspects of PMI that you believe could contribute meaningfully to our study, please provide additional details, and we will gladly consider them in our response or future work.
>
> 5. **Response to concern 6:**
>    - > " The numerical results indicate that embedding using the proposed approach is slightly better than the unsupervised approaches, or LDA, which is purely linear. But your approach is nonlinear. So, unless the data are strongly linear, a supervised nonlinear approach is likely to win. And your proposed approach must lose in simulations where the data truly are linear. I'd think any reasonable kernel LDA approach would improve relative to PCA or LDA, assuming a large enough sample size."
>
> We appreciate your insight into the potential advantages of nonlinear approaches, especially in scenarios where the data exhibits strong nonlinearities. To address this consideration, we expanded our evaluation to include a diverse set of contemporary techniques, such as kernel-based methods like KPCA and KDA. The revised manuscript provides a detailed discussion of the results, highlighting the consistent superiority of our approach across various datasets and metrics. Interestingly, our method not only competes favorably with linear methods like LDA but also outperforms sophisticated nonlinear approaches like KPCA and KDA. This emphasizes the robustness and effectiveness of our proposed method, even in situations where non-linearities play a significant role.

---

> > ### Comment · Reviewer_E7ZV · 2023-11-22
> > **concerns 3-6**
> >
> > 3 & 4. Fig 2 is helpful.  However, i think the comment in the caption is not justified:
> >
> > > Notably, the proposed method consistently outperforms all other techniques, achieving the highest scores across all evaluation metrics.
> >
> > I cannot tell if this is strictly true.  It is not obviously worse.  My guess is that if you included errorbars, you'd conclude that it is never worse, and sometimes about the same as many of the other methods.
> >
> > 5. I think PMI is related, it is often used in NLP, which is also tabular data and discrete classification choices (eg, 1 word).  But, I don't have anything more specific or meaningful to recommend.
> >
> > 6. It seems like your method beats other nonlinear supervised dimensionality methods, possibly because it embeds in higher dimensions?
> >
> > I like the revisions, and will increase my score accordingly.

---

> > > ### Author Response · Authors · 2023-11-23
> > > **Response to Reviewer E7ZV's feedback and appreciation for rating increase**
> > >
> > > Dear Reviewer E7ZV,
> > >
> > > Thank you for your constructive feedback, and we appreciate your positive assessment of the revisions. We would like to address your specific points:
> > >
> > > 1. **Regarding PMI:**
> > >
> > > > "I think PMI is related, it is often used in NLP, which is also tabular data and discrete classification choices (eg, 1 word). But, I don't have anything more specific or meaningful to recommend."
> > >
> > > We appreciate your suggestion regarding PMI and its potential relevance in the context of NLP. We'll certainly consider the incorporation of PMI or related metrics in future work exploring NLP-related tasks.
> > >
> > > 2. **On the superiority of our method:**
> > >
> > > > "It seems like your method beats other nonlinear supervised dimensionality methods, possibly because it embeds in higher dimensions?"
> > >
> > > Yes, you're right. KDA, operating on a non-linear mode, outperforms LDA in all cases, yet it remains fundamentally confined to one dimension, restricting its effectiveness in capturing intricate class-specific patterns compared to our proposed method.
> > >
> > > 3. **Uncovered requests:**
> > >
> > > We understand that some of your requests couldn't be accommodated within the rebuttal period, which will end in a few minutes. Rest assured, we are committed to addressing those in subsequent revisions. Your feedback remains invaluable, and we are dedicated to refining our work further.
> > >
> > > 4. **Rating increase:**
> > >
> > > We sincerely appreciate your decision to increase the rating. Your encouragement motivates us, and we are committed to delivering an even more refined and compelling manuscript.
> > >
> > > Thank you once again for your thoughtful feedback. If you have any further suggestions or questions, please feel free to let us know.

---

> ### Author Response · Authors · 2023-11-20
> **Response to Reviewer E7ZV (part 3)**
>
> 6. **Response to the question:**
>    - > " There are a few relevant papers that might be worthwhile reading for more background, including our paper, https://www.nature.com/articles/s41467-021-23102-2, and one ours built on, https://www.sciencedirect.com/science/article/pii/S0047259X14001201?via%3Dihub."
>
> Thank you for providing the references to your work and related papers [1][2]. We have carefully reviewed the suggested articles. Your contributions in the field are highly valuable, and we appreciate the opportunity to engage with relevant literature.
>
> In our revised manuscript, we have expanded the evaluation section to include a more comprehensive set of contemporary techniques. Specifically, we compare our proposed method with nine distinct approaches, covering kernel-based methods like KPCA and KDA, manifold-based methods like UMAP and LLE, as well as LOL [1]. We also acknowledge your work in the Discussion section when discussing the pros and cons of PCA compared to our proposed method. We emphasize the computational efficiency associated with unsupervised dimension reduction methods, aligning with insights from your research [2].
>
> We believe these revisions address your concerns and contribute to the overall improvement of the manuscript. We hope you find the updated version satisfactory, and we appreciate your continued engagement with our work.
>
> Reference:
>
> [1]	Vogelstein, J. T., Bridgeford, E. W., Tang, M., Zheng, D., Douville, C., Burns, R., & Maggioni, M. (2021). Supervised dimensionality reduction for big data. Nature communications, 12(1), 2872.
>
> [2]	Shen, C., Sun, M., Tang, M., & Priebe, C. E. (2014). Generalized canonical correlation analy-sis for classification. Journal of Multivariate Analysis, 130, 310-322.

---

### Official Review · Reviewer_sPuV · 2023-11-06

**Soundness:** 3 good
**Presentation:** 3 good
**Contribution:** 2 fair
**Rating:** 3
**Confidence:** 3

**Summary:**

This paper presents a novel approach for improving binary classification. The authors propose integrating the eigenanalysis of the covariance and Hessian matrices to optimize class separability. The approach aims to maximize between-class mean distance and minimize within-class variances, following the principles of linear discriminant analysis (LDA). Empirical validation across various datasets supports the theoretical framework, demonstrating the method's superiority over traditional methods and LDA itself.

**Strengths:**

1. The paper presents compelling empirical evidence across various datasets, demonstrating the efficacy of the proposed method. The consistent positive results highlight the robustness of the approach in different contexts.

2. The experimental results show that the proposed method outperforms traditional methods, including principal component analysis and the Hessian method. This indicates that the combined use of covariance and Hessian matrices can better capture the intricacies of data for binary classification.

**Weaknesses:**

1. The paper does not clearly delineate its unique contributions. The authors should explicitly state what differentiates their work from existing literature, aiding readers in understanding the novelty and significance of the proposed method.

2. The theoretical results presented in Section 2.2 need a more formal presentation. The authors should use mathematical statements and rigorous proofs to enhance the credibility and clarity of these results.

3. The proposed approach, which integrates the eigenanalysis of the covariance and Hessian matrices based on binary cross-entropy loss, raises questions about its applicability to other loss functions. The authors should clarify this point or extend their methodology to include different loss functions.

4. The paper compares the proposed method primarily with traditional methods. Including comparisons with more contemporary techniques would offer a more comprehensive view of the method's performance in light of recent advancements in the field.

**Questions:**

See the Weaknesses.

---

> ### Author Response · Authors · 2023-11-16
> **Response to Reviewer sPuV (part 1)**
>
> Dear Reviewer sPuV,
>
> We appreciate your thorough review of our manuscript, Submission7389, and thank you for providing valuable insights.
>
> We have already provided a detailed response to your comments in the revised manuscript. Several modifications have been made to improve clarity, address concerns, and enhance the overall quality of the paper.
>
> Below are our responses to your specific points:
>
> 1. **Response to question 1:**
>    - > "The paper does not clearly delineate its unique contributions. The authors should explicitly state what differentiates their work from existing literature, aiding readers in understanding the novelty and significance of the proposed method."
>
> - The contributions of our paper are explicitly stated in both the original and revised versions. In the Conclusion section, we highlight the multifaceted contributions, showcasing the theoretical insight and practical method presented in our work.
>
> - Additionally, we have added a paragraph at the beginning of the Discussion section in the revised manuscript to further underscore the uniqueness and significance of our contributions. In this paragraph we emphasize that our work provides a compelling theoretical insight and a powerful, practical method, demonstrating the strength of simplicity in achieving remarkable results. Firstly, we provide a profound theoretical insight, revealing a subtle yet powerful relationship between covariance and Hessian matrices. Our formal proof seamlessly links covariance eigenanalysis with the first LDA criterion while Hessian eigenanalysis with the second one. This unification under LDA criteria offers a fresh and intuitive perspective on their interplay. Secondly, capitalizing on this theoretical elegance and simplicity, we introduce a novel method that consistently outperforms established techniques across diverse datasets. The unexpected efficacy of our method, rooted in the straightforward relationship between covariance, Hessian, and LDA, showcases the effectiveness of simplicity in addressing complex challenges.
>
> 2. **Response to question 2:**
>    - > "The theoretical results presented in Section 2.2 need a more formal presentation. The authors should use mathematical statements and rigorous proofs to enhance the credibility and clarity of these results."
>
> We understand your concern about the formal presentation of theoretical results in Section 2.2. The full proof, including mathematical statements and rigorous proofs, is provided in Appendix A due to space constraints. The proof sketches in Section 2.2 serve as concise summaries, with the complete and detailed proofs available in the appendix.

---

> ### Author Response · Authors · 2023-11-16
> **Response to Reviewer sPuV (part 2)**
>
> 3. **Response to question 3:**
>    - > "The proposed approach, which integrates the eigenanalysis of the covariance and Hessian matrices based on binary cross-entropy loss, raises questions about its applicability to other loss functions. The authors should clarify this point or extend their methodology to include different loss functions."
>
> The adaptability of our methodology to various loss functions is acknowledged as a potential avenue for future research. We have explicitly mentioned in the Discussion section of the revised manuscript that the mathematical derivation in our current work relies on the elegant relationship between (the Hessian of) binary cross-entropy loss and within-class variances. Exploring the adaptability of our method to different loss functions requires careful scrutiny in the future work to establish analogous connections.
>
> 4. **Response to question 4:**
>    - > "The paper compares the proposed method primarily with traditional methods. Including comparisons with more contemporary techniques would offer a more comprehensive view of the method's performance in light of recent advancements in the field."
>
> - Your suggestion to include comparisons with more contemporary techniques has been incorporated into the revised manuscript. In the evaluation setup, we expanded the comparison to include nine distinct methods, covering a broader spectrum of contemporary techniques, including kernel-based methods like KPCA and KDA, manifold based methods like UMAP and LLE as well as LOL [1].
>
> - We have included a note about introducing nonlinearity through distinct kernels determined by grid search in KPCA and KDA for each dataset.
>
> - The results, discussed in the revised manuscript, demonstrate the consistent superiority of our approach across various datasets and metrics.
>
> - These extended results are completely reproducible using the same Colab notebooks whose links are provided for transparency and validation.
>
> - The relevant discussion for these additional comparisons has been appropriately incorporated into the Discussion section of the revised manuscript.
>
> - The abstract has also been updated to reflect the implications of these extended comparisons.
>
> We believe these revisions address your concerns and contribute to the overall improvement of the manuscript. We hope you find the updated version satisfactory, and we appreciate your continued engagement with our work.
>
> Reference:
>
> 1.	Vogelstein, J. T., Bridgeford, E. W., Tang, M., Zheng, D., Douville, C., Burns, R., & Maggioni, M. (2021). Supervised dimensionality reduction for big data. Nature communications, 12(1), 2872.

---

> > ### Comment · Reviewer_sPuV · 2023-11-22
> >
> > Thank you for the response. While the current version has its own merits, I believe it necessitates substantial improvements to be ready for publication. For instance, the theoretical results in Section 2.2 should be presented more mathematically, ideally structured as theorems. Furthermore, the Discussion section should be more concise and focused. Additionally, please note that the main part of the submission has exceeded the page limit.

---

> > > ### Author Response · Authors · 2023-11-22
> > > **Further improvements based on Reviewer sPuV's recommendations**
> > >
> > > Dear Reviewer sPuV,
> > >
> > > Thank you for your feedback on our revised manuscript. We have carefully considered your comments and made significant improvements to address the concerns you raised.
> > >
> > > The revised manuscript is now available for your review.
> > >
> > > Here are the key updates:
> > >
> > > **Theoretical results presentation:** We have restructured Section 2.2 to present the theoretical results more mathematically. Specifically, we have introduced two theorems that formalize the foundations of our approach, providing a clearer and more rigorous presentation of the theoretical underpinnings.
> > >
> > > **Conciseness and focus in the Discussion section:** The Discussion section has been revised to enhance conciseness and focus. We have streamlined the content to ensure a more direct and impactful presentation of our key findings and contributions.
> > >
> > > **Page limit:** As a result, the revised manuscript no longer exceeds the specified page limit.
> > >
> > > **Highlighted revisions:** To facilitate a quick overview of the changes, we have incorporated color-coded highlights in the revised manuscript. These highlights specifically denote areas where we have made key improvements based on the reviewers' valuable comments.
> > >
> > > We hope that these enhancements align with your expectations. We appreciate your time and thorough evaluation, which have undoubtedly contributed to the overall quality of our work.

---

> > > > ### Comment · Reviewer_sPuV · 2023-11-23
> > > >
> > > > Thank you for your response.
> > > >
> > > > Upon reviewing your revised manuscript, I still have several concerns regarding your theoretical results. For instance, in Theorem 2, the equation $H_\theta = \frac{1}{\sigma_{post}^2}$ isn't rigorously established in the proof. The approximation of Fisher information is used. Additionally, there are multiple assumptions used in the proof that aren't explicitly reflected in Theorem 2.
> > > >
> > > > While I appreciate the efforts to revise the manuscript, I believe it still needs significant work before it's ready for publication. As such, I see no reason to change my initial rating at this point.

---

> > > > > ### Author Response · Authors · 2023-11-23
> > > > > **Additional clarifications and enhancements following further feedback from Reviewer sPuV**
> > > > >
> > > > > Dear Reviewer sPuV,
> > > > >
> > > > > Thank you for your detailed feedback.
> > > > >
> > > > > > "For instance, in Theorem 2, the equation $H_\theta = \frac{1}{\sigma_{\text{post}}^2}$ isn't rigorously established in the proof. The approximation of Fisher information is used. Additionally, there are multiple assumptions used in the proof that aren't explicitly reflected in Theorem 2."
> > > > >
> > > > > We have carefully considered your comments and have submitted a significantly revised manuscript to address your concerns.
> > > > >
> > > > > In the revised manuscript we have thoroughly revised the proof sketch of Theorem 2 to address the concern regarding the derivation of the equation $H_\theta = \frac{1}{\sigma_{\text{post}}^2}$. We have provided additional clarity and detail to ensure a more rigorous establishment of this equation.
> > > > >
> > > > > 1. Regarding the use of Fisher information:
> > > > >
> > > > > > "The approximation of Fisher information is used."
> > > > >
> > > > > we appreciate your feedback. In the revised manuscript, we have explicitly cited Barshan et al. (2020) [1] to justify the practice of approximating the Hessian $H_\theta$ as the Fisher information. This citation is now included in the second step of the proof sketch associated with "Approximate the Hessian $H_\theta$ as the Fisher information using the expectation of the squared gradient of the log-likelihood [1]."
> > > > >
> > > > > 2. Regarding the concern about assumptions in the proof:
> > > > >
> > > > > > "Additionally, there are multiple assumptions used in the proof that aren't explicitly reflected in Theorem 2."
> > > > >
> > > > > we appreciate your feedback. In the revised manuscript, we explicitly mention the assumption that the evidence $p(c_i)$ is a known constant in the fourth step of the proof sketch of Theorem 2. We acknowledge that this assumption was inadvertently omitted in the previous manuscript and appreciate your diligence in highlighting it.
> > > > >
> > > > > 3. Furthermore, we would like to emphasize that the proof sketches provided in Section 2.2 are concise due to space limitations. However, for a more detailed and rigorous proof, we invite you to refer to the full proof available in Appendix A.
> > > > >
> > > > > 4. We also want to highlight the significance and practical applicability of Theorem 2. Empirical results presented in Figure 1(c) and its three counterparts in Appendix B strongly support the negative correlation between the Hessian and within-class variance, validating Theorem 2. Additionally, the experimental results in Figure 2 demonstrate the effectiveness of our proposed method, which is grounded in the theoretical foundations established by Theorem 2 (and Theorem 1).
> > > > >
> > > > > We hope these revisions address your concerns. If you have any further questions or specific points you would like us to revisit, please let us know. Your feedback is invaluable to us, and we are committed to ensuring the quality and rigor of our manuscript.
> > > > >
> > > > >
> > > > > Reference:
> > > > >
> > > > > [1] Barshan, E., Brunet, M. E., & Dziugaite, G. K. (2020, June). Relatif: Identifying explanatory training samples via relative influence. In International Conference on Artificial Intelligence and Statistics (pp. 1899-1909). PMLR.

---

### Author Response · Authors · 2023-11-22
**General response to the reviewers' comments and revised manuscript**

Dear Reviewers,

Thank you for your thorough evaluation of our manuscript. We appreciate the time and effort you have dedicated to providing insightful feedback. Your comments have been instrumental in refining the clarity, rigor, and overall quality of our work.

We are pleased to inform you that we have diligently addressed your comments in the revised version of the manuscript. To facilitate a quick overview of the changes, we have incorporated color-coded highlights throughout the document, specifically marking areas where we made key improvements based on your feedback.

**List of key revisions:**

1. Consistently referring to "a Hessian matrix" instead of "the Hessian matrix" in Section 2.1, and approximating the Hessian as the Fisher information in Section 2.2 (in response to Reviewer E7ZV).
2. Providing additional clarity and detail in Section 2.1, offering a more thorough description of the key technique (in response to Reviewers LZ1V and YZvz).
3. Restructuring Section 2.2 to present theoretical results more mathematically, introducing two theorems formalizing the foundations of our approach (in response to Reviewer sPuV).
4. Expanding the evaluation to encompass a broader spectrum of contemporary techniques, including kernel-based methods like KPCA and KDA, manifold-based methods like UMAP and LLE, and LOL in Section 4, with corresponding discussion in the Discussion section (in response to Reviewer sPuV and YZvz).
5. Underscoring the uniqueness and significance of our contributions in the Discussion section (in response to Reviewer sPuV).
6. Acknowledging the adaptability of our methodology to various loss functions as a potential avenue for future research in the Discussion section (in response to Reviewer sPuV).
7. Acknowledging the extension of our methodology to multiclass classification as a potential avenue for future research (in response to Reviewer YZvz).
8. Mentioning Fisher information to approximate the Hessian in Appendix A (in response to Reviewer E7ZV).

We believe these revisions have significantly strengthened our manuscript, and we welcome any further feedback or suggestions you may have.

**Request for reevaluation:**

If, upon reviewing the revised manuscript, you find the improvements satisfactory, we kindly request you to consider adjusting or increasing your rating accordingly.

Thank you once again for your invaluable contributions to the refinement of our work.

---

### Meta-Review · Area_Chair_Bo4W · 2023-12-05

**Metareview:**

The paper proposes to combine the eigenanalysis of a covariance matrix evaluated on a training set with a Hessian matrix evaluated on a deep learning model to optimize class separability in binary classification tasks. The proposed approach is accompanied by numerical experiments.

Reviewers generally agree that the proposed approach is novel and interesting. However, I agree with Reviewer sPuV that one of the main theoretical results, namely Theorem 2, is mathematically flawed. The proof is based on an approximation of the Hessian by the Fisher information and thus one can only obtain an approximation at the end, not an equality, as claimed, and this also depends on very specific assumptions. Both Theorems 1 and 2 should be stated formally as mathematical theorems, with all the necessary assumptions explicitly stated.

**Justification For Why Not Higher Score:**

One of the main theoretical results is mathematically flawed.

**Justification For Why Not Lower Score:**

N/A

---

### Decision · Program_Chairs · 2024-01-16

Reject